# COVID-19-Related Stress and Anxiety, Body Mass Index, Eating Disorder Symptomatology, and Body Image in Women from Poland: A Cluster Analysis Approach

**DOI:** 10.3390/nu13041384

**Published:** 2021-04-20

**Authors:** Kamila Czepczor-Bernat, Viren Swami, Adriana Modrzejewska, Justyna Modrzejewska

**Affiliations:** 1Institute of Psychology, University of Wroclaw, Dawida 1, 50-527 Wroclaw, Poland; 2School of Psychology and Sport Science, Anglia Ruskin University, East Road, Cambridge CB1 1PT, UK; viren.swami@aru.ac.uk; 3Centre for Psychological Medicine, Perdana University, Changkat Semantan, Kuala Lumpur 50490, Malaysia; 4Department of Psychology, Katowice Business University, Harcerzy Września 1939 3, 40-659 Katowice, Poland; adriana.modrzejewska@gwsh.pl; 5Institute of Pedagogy, University of Bielsko-Biala, Willowa 2, 43-309 Bielsko-Biala, Poland; jmodrzejewska@ath.bielsko.pl

**Keywords:** eating disorders, body image, COVID-19, anxiety, stress, body weight, women

## Abstract

To limit the spread of the novel coronavirus (COVID-19), many countries have introduced mandated lockdown or social distancing measures. Although these measures may be successful against COVID-19 transmission, the pandemic and attendant restrictions are a source of chronic and severe stress and anxiety which may contribute to the emergence or worsening of symptoms of eating disorders and the development of negative body image. Therefore, in this study, we aimed to: (1) classify different conditions associated with COVID-19-related stress, COVID-19-related anxiety, and weight status; and (2) analyze and compare the severity of dimensions typically related to eating disorders symptomatology and body image in individuals with different COVID-19-related stress, COVID-19-related anxiety, and weight status. Polish women (*N* = 671, *M*_age_ = 32.50 ± 11.38) completed measures of COVID-19-related stress and anxiety along with body dissatisfaction, drive for thinness, and bulimia symptomatology subscales of the Eating Disorders Inventory, and the appearance evaluation, overweight preoccupation, and body areas satisfaction subscales of the Multidimensional Body-Self Relations Questionnaire. The following four clusters were identified through cluster analysis: (a) Cluster 1 (*N* = 269), healthy body weight and low COVID-related stress (*M* = 3.06) and anxiety (*M* = 2.96); (b) Cluster 2 (*N* = 154), healthy body weight and high COVID-related stress (*M* = 5.43) and anxiety (*M* = 5.29); (c) Cluster 3 (*N* = 127), excess body weight and high COVID-related stress (*M* = 5.23) and anxiety (*M* = 5.35); (d) Cluster 4 (*N* = 121), excess body weight and low COVID-related stress (*M* = 2.69) and anxiety (*M* = 2.83). Our results showed that Clusters 3 and 4 had significantly greater body dissatisfaction and lower appearance evaluation and body areas satisfaction than Clusters 1 and 2. Cluster 3 also had a significantly higher level of drive for thinness, bulimia, and overweight preoccupation than Clusters 1 and 2. These preliminary findings may mean that the COVID-19 pandemic and attendant anxiety and stress caused by the pandemic are exacerbating symptoms of eating disorders and negative body image, with women with excess weight particularly at risk.

## 1. Introduction

To limit the spread of the novel coronavirus (COVID-19), many countries have introduced mandated lockdown or social and physical distancing measures (e.g., travel restrictions; leaving home only for food, work, and health reasons; remote working; and closure of health services) that have led to significant changes in daily living [1,2,3]. Although these prevention measures may be successful against COVID-19 transmission, the pandemic and attendant restrictions have also been a source of chronic and severe stress and anxiety [4,5,6,7,8,9]. Indeed, evidence suggests that the pandemic has contributed to the deterioration of mental health in populations worldwide, including the emergence or worsening of symptoms of eating disorders and the development of negative body image [2,10,11,12,13,14].

Increased stress and anxiety related to COVID-19, leading to a worsening in eating disorder symptoms and body image, may have numerous causes, including the following: (1) changes to daily routine, including limited access to leisure facilities and gyms, which may escalate concerns about one’s own body [2,10,11,12,14,15]; (2) increased media coverage of food, its purchase, and potential shortages, which may increase ruminations about eating or promoting stockpiling and binging [2,10,12]; (3) persistent exposure to COVID-19-related information in the media (e.g., number of deaths and infections), which may increase concerns about one’s own health (especially in immunocompromised groups, e.g., patients with anorexia) [12,16,17,18,19,20]; (4) changes in the performance of professional duties, which have been largely related to the need to work remotely and contact colleagues through videoconferencing suites (increased number of situations in which there is exposure and the need to watch oneself and view one’s own appearance, and the need to learn new skills related to self-presentation) [2,21]; (5) increased feelings of losing control over the current situation, which may trigger behaviors aimed at “regaining control through body control” [14,15]; (6) increased time spent on social media, which may exacerbate concerns about appearance and one’s body and lead to an increase in compensatory behavior (e.g., restrictive diet and vomiting) [2,12,22]; (7) limited contact with other people, which may be associated with the legitimization of some eating disorder symptoms (e.g., avoidance of social eating and body exposure) [12,15]; (8) disruption of access to specialist support and treatment service [2,15]; (9) limited possibility of direct contact with family, friends, and other people, who are important sources of daily support and help to regulate emotions through adaptive mechanisms [23,24].

Although knowledge about the symptoms of eating disorders and negative body image during the COVID-19 pandemic has increased in diverse populations [13,24,25,26,27,28,29,30], research and knowledge on the symptoms of eating disorders and body image in Polish women remains limited (Polish studies published so far have concerned either the period before the pandemic [31] or a general problem of nutrition behaviors before and during COVID-19 lockdown [32]). Given the likelihood that the pandemic, attendant restrictions, and their consequences have been a source of chronic and severe stress and anxiety [4,5,6,7,8,9], and, as earlier research indicated, may have had an impact on body image and eating disorder symptomatology [10,13,14], it is vital to develop a fuller understanding of this issue in Polish women. Moreover, clarity on this issue is essential to ensure that appropriate interventions are made, which will be able to take into account the specific situation related to COVID-19 in Poland. Therefore, this work aimed to: (1) classify different conditions associated with COVID-19-related stress, COVID-19-related anxiety, and weight status; and (2) analyze and compare the severity of dimensions typically related to eating disorders symptomatology and body image in individuals with different COVID-19-related stress, COVID-19-related anxiety, and weight status. We hypothesized that women with excess body weight, high anxiety, and high stress related to COVID-19 would have significantly greater symptoms of eating disorders and negative body image than those of healthy weight, low anxiety, and low stress.

## 2. Materials and Methods

### 2.1. Participants and Procedure

This study was approved by the Ethics Committee of the University of Wroclaw (14 December 2020) and conducted in accordance with the Declaration of Helsinki. Data were collected via an online survey (populated on Google Forms) between 14 December 2020 and 1 February 2021 (for most of that time, there was a total lockdown in Poland, except for a short period before Christmas). Participants were recruited via flyers (posted at universities and workplace locations) and an online advertisement. After contacting the researchers, all women provided prior written informed consent and were made aware that participation in our study was voluntary and anonymous. Next, they completed an online survey, which was accessed via a link sent to participants. All participation was voluntary, and participants were not remunerated.

Seven hundred and forty-seven women volunteered for the study, 76 of whom were outliers, and therefore were removed from the database. Our final sample included 671 Polish women who ranged in age from 18 to 73 years (*M* = 32.50, *SD* = 11.38) and in self-reported body mass index (BMI) from 18.52 to 50.70 kg/m^2^ (*M* = 24.78, *SD* = 5.10). The majority of participants were White (99.1%) and heterosexual (91.2%). In terms of relationship status, 17.6% were single, 35.3% were partnered but not married, 41.6% were married, and 5.5% had some other status. In terms of education, 27.7% had completed secondary or technical school, 17.9% had completed an undergraduate degree, 32.8% had completed a Master’s degree, 14% had completed another postgraduate degree, and 7.60% had completed some other qualification.

### 2.2. Measures

Five measures were included in the online survey. We obtained permission to use all instruments from their developers. For instruments where the Polish version was not available, we produced novel Polish translation using the standardized back-translation procedure [33].

#### 2.2.1. COVID-19-Related Stress

To measure levels of stress caused by the COVID-19 pandemic, we used a 5-item measure, which a previous study had shown to have adequate construct validity [13]. Participants were asked how stressed they felt about the impact of the COVID-19 pandemic on their daily lives, finances, personal relationships, work and/or studies, and their future in general. All items were rated on a 7-point scale ranging from 1 (*not at all stressed*) to 7 (*extremely stressed*), with higher mean scores reflecting greater COVID-19-related stress. This instrument has not been previously used in Polish-speaking populations. Therefore, we examined the factor structure of scores on the instrument using a principal-axis exploratory factor analysis. The results supported the extraction of a single factor consisting of all 5 items (KMO = 0.83, Bartlett’s test of sphericity *χ*^2^_(10)_ = 2042.35, *p* < 0.001, eigenvalue = 3.42, 68.4% of the variance explained, item-factor loadings = 0.79 to 0.86). On the basis of the results, we computed an overall score as the mean of all five items. McDonald’s ω for scores on this measure was 0.89.

#### 2.2.2. COVID-19-Related Anxiety

To measure anxiety caused by the COVID-19 pandemic, we used a 1-item measure in which we asked participants how anxious they are about the COVID-19 pandemic [8,13]. In previous studies, scores on this measure have been shown to have adequate construct and predictive validity [8,13]. Items were rated on a 7-point scale (from 1 = *not anxious* at all to 7 = *extremely anxious*). The higher the score, the greater the COVID-19-related anxiety.

#### 2.2.3. Eating Disorders Symptoms

To measure symptoms of eating disorders, participants completed the Eating Disorder Inventory (EDI), which has adequate construct and factorial validity, including in Polish women [34,35]. The EDI consists of the following 8 subscales: body dissatisfaction, drive for thinness, bulimia, ineffectiveness, perfectionism, interpersonal distrust, interoceptive awareness, and maturity fears. In this study we used the scores of three of the subscales, i.e., body dissatisfaction (EDI-BD), drive for thinness (EDI-DT), and bulimia (EDI-B). Mean scores were computed so that higher scores reflected greater body dissatisfaction, drive for thinness, and bulimia symptoms. All items were rated on a 6-point scale ranging from 0 (*never*) to 5 (*always*). After inverting the indicated items, responses were recoded according to previous recommendations (*always* = 3, *usually* = 2, *often* = 1, *sometimes*, *rarely*, *never* = 0). McDonald’s ω for EDI-BD scores was 0.88, for EDI-DT scores was 0.86, and for EDI-B scores was 0.88.

#### 2.2.4. Body Image

To measure body image, participants were asked to complete the Multidimensional Body–Self Relations Questionnaire (MBSRQ), which has been shown to have adequate construct and factorial validity, including in Polish women [36,37]. The MBSRQ includes the following 9 subscales measuring different body image-related aspects: appearance evaluation, fitness evaluation, health evaluation, appearance orientation, fitness orientation, health orientation, overweight preoccupation, self-classified weight, and body areas satisfaction. In this study, we used the scores of three of the subscales, i.e., appearance evaluation (MBSRQ-AE), overweight preoccupation (MBSRQ-OWP), and body areas satisfaction (MBSRQ-BASS). Mean scores were computed so that higher scores reflected higher satisfaction with the body as a whole and positive feelings related to them (AE), higher fat-related anxiety, food restriction, and weight vigilance (OWP), and higher satisfaction with individual parts of the body (BASS). Items were rated on variable 5-point scales. McDonald’s ω for MBSRQ-AE scores was 0.90, for MBSRQ-OWP scores was 0.70, and for MBSRQ-BASS scores was 0.86.

#### 2.2.5. Demographics

Participants were asked to provide their demographic information (age, ethnicity, sexual orientation, educational qualification, and relationship status), weight and height. We calculated BMI (kg/m^2^) based on self-report data.

### 2.3. Statistical Analysis

Our analyses were conducted using IBM SPSS Statistic version 26. A two-step cluster analysis (with Schwarz’s Bayesian criterion, BIC) was used to identify clusters based on COVID-19-related stress, COVID-19-related anxiety, and BMI. This was a hybrid approach based on a combination of running a distance measure (to separate groups) and hierarchical methods (to select the optimal subgroups model). A two-step cluster analysis was chosen as it is appropriate for both categorical and continuous variables and samples of *N* > 200 [38]. Multivariate analysis of variance (MANOVA) was used to assess differences between the clusters with regard to eating disorder symptoms (body dissatisfaction, drive for thinness, and bulimia), and body image (appearance evaluation, overweight preoccupation, and body areas satisfaction). To correct for multiple comparisons, the Bonferroni corrected/adjusted *p*-value was used. A 5% significance level was used.

## 3. Results

### 3.1. Cluster Analysis of COVID-19-Related Stress, COVID-19-Related Anxiety, and Body Mass Index

The four clusters were labeled, and were characterized, as follows: (a) Cluster 1 (*N* = 269), healthy body weight and low COVID-related stress (*M* = 3.06) and anxiety (*M* = 2.96); (b) Cluster 2 (*N* = 154), healthy body weight and high COVID-related stress (*M* = 5.43) and anxiety (*M* = 5.29); (c) Cluster 3 (*N* = 127), excess body weight and high COVID-related stress (*M* = 5.23) and anxiety (*M* = 5.35); (d) Cluster 4 (*N* = 121), excess body weight and low COVID-related stress (*M* = 2.69) and anxiety (*M* = 2.83). Table 1 presents the demographic characteristics of these clusters.

### 3.2. Comparison of the Four Clusters for Eating Disorder Symptoms and Body Image

Using Pillai’s trace, a multivariate analysis of variance (MANOVA) indicated a significant effect of clusters on all eating disorders symptoms, *V* = 0.16, *F*(3, 665) = 12.77, *p* < 0.001, and body image variables, *V* = 0.19, *F*(3, 665) = 15.01, *p* < 0.001. Table 2 shows results with Bonferroni’s adjustment for multiple comparisons.

Our results indicated that Clusters 3 and 4 did not significantly different from each other in terms of all variables. With regard to eating disorder symptomatology, both of the abovementioned clusters had significantly greater body dissatisfaction than Clusters 1 and 2. Cluster 3 also had a higher level of drive for thinness and bulimia than Clusters 1 and 2. In turn, Cluster 4 differed significantly only from Cluster 1. In relation to body image, Clusters 3 and 4 had significantly lower appearance evaluation and body areas satisfaction than Clusters 1 and 2. Cluster 3 also had a higher level of overweight preoccupation than Clusters 1 and 2. In turn, Cluster 4 differed significantly only from Cluster 1.

Overall, our results partially support our hypothesis, as higher levels of symptoms of eating disorders and negative body image were observed in women with excess body weight, high anxiety, and stress related to COVID-19 as compared with women with a healthy body weight and with low levels of anxiety and stress. Our results indicate that both COVID-19-related stress and anxiety, as well as body weight status, may contribute to the intensity of eating disorder symptomatology and negative body image.

## 4. Discussion

This study examined patterns of COVID-19-related stress, COVID-19-related anxiety, and weight status in a sample of Polish women during the COVID-19 pandemic. Four clusters were uncovered. Cluster 3 (excess body weight, high COVID-19-related stress, and high COVID-19-related anxiety)–was associated with significantly higher levels of both eating disorder symptoms and negative body image than Clusters 1 (healthy body weight, low COVID-19-related stress, low COVID-19-related anxiety) and 2 (healthy body weight, high COVID-19-related stress, high COVID-19-related anxiety). More specifically, our results show that women from Cluster 3 had significantly higher levels of body dissatisfaction, drive for thinness, bulimia, overweight preoccupation, and lower levels of appearance evaluation and body areas satisfaction than Clusters 1 and 2, although this cluster did not differ significantly from Cluster 4 (excess body weight, low COVID-19-related stress, and low COVID-19-related anxiety) in any dimension. Moreover, although Cluster 4 also differed significantly from Cluster 1 in all dimensions, such differences were not always significant between Cluster 4 and Cluster 2 (drive for thinness, bulimia, overweight preoccupation). This may mean that for such aspects of functioning such as drive for thinness, bulimia symptoms, and overweight preoccupation (fat-related anxiety, food restriction and weight vigilance), a risk factor for their high level is, in addition to excess body weight, high levels of COVID-19-related stress and anxiety. For the rest of the assessed body-related assessment and feelings aspects (body dissatisfaction, appearance evaluation, and body areas satisfaction), it seems to be actually the condition of excess weight irrespective of COVID-19-related stress and anxiety.

Our results partially support our hypothesis, as higher levels of symptoms of eating disorders and negative body image were observed in women with excess body weight, high anxiety, and stress related to COVID-19 as compared with women with a healthy body weight with low levels of anxiety and stress. Therefore, it is possible that our research reflects what earlier research has highlighted in other national contexts, namely that the COVID-19 pandemic, particularly the stress and anxiety caused by the pandemic, are having a detrimental impact on mental health, including in terms of eating behaviors and body image [4,5,6,7,8,9,25,28]. Individuals may response to such stress and anxiety by employing a range of maladaptive strategies, such as behaviors that exacerbate body image concerns (e.g., body ruminations and body checking) and disordered eating symptoms (e.g., restrictive eating). In this sense, higher levels of COVID-19-related stress and anxiety appear to contribute to a greater intensity of eating disorder symptomatology and a negative body image, which is consistent with previous work [10,12,13,24,28], although our results suggest a degree of nuance in this finding.

Specifically, our results suggest that excess weight, as indicated by BMI, is associated with higher levels of body dissatisfaction [13], and also higher levels of other symptoms of eating disorders. As mentioned by Phillipou et al. [25], COVID-19 and attendant restrictions, stress, and anxiety may lead to an increase in restrictive and binge eating behaviors, which is also consistent with our study, since the presence and severity of these behaviors were analyzed within the drive for thinness (EDI-DT) and bulimia (EDI-B) subscales. This is also reflected in other studies that have highlighted the effect of COVID-19 on drive for thinness [29] and bulimia symptoms (binge eating and self-induced vomiting) [27,28]. For instance, one German study showed that, due to the COVID-19 pandemic, the quality of life of patients had significantly decreased, levels of depression had increased, and the frequency of using face-to-face psychotherapy had decreased [27], which, as we know, may additionally strengthen the experience of anxiety and stress.

The present results indicate that both stress and anxiety associated with the current pandemic, as well as body weight, can be included in the further analysis of levels of intensity of eating disorder symptoms and negative body image. However, it is necessary to interpret our results with caution, as they could be determined primarily by body weight. It is possible that, especially among women with excess body weight, the triggers and changes related to COVID-19, already mentioned in the Introduction (e.g., increasing feeling of loss control over the current situation, reduced use of gyms, increased screen time, and exposure to thin ideals via social media), may intensify the experience of negative emotions and concerns about weight or shape changes [2,10,29,39]. Moreover, although we have not assessed causal relationships, it may be assumed that increased stress, anxiety, and concerns are a source of body dissatisfaction, which is accompanied by constant vigilance and monitoring of weight or body shape [10,12,40,41]. The COVID-19 pandemic has caused many changes in daily lives, and these changes have also been associated with less physical activity and more unhealthy eating behaviors [26,42,43,44]. These changes may contribute to the risk of weight gain becoming greater and body assessment more negative [26,43,44,45]. However, women who want to control these body changes and want to reduce increased weight-related anxiety may turn to measures that still remain available, including food restrictions, vomiting, and other compensatory methods [46,47].

The preliminary reports from various countries may indeed mean that the COVID-19 pandemic is exacerbating symptoms of eating disorders and negative body image and that women with excess body weight are particularly at risk. This also seems to be confirmed by the data which show that over 30% of patients suffering from eating disorders experience an exacerbation of symptoms, and institutions caring for patients with eating disorders report the need to increase the scope of support [2,11,48,49]. This crisis situation, which is the COVID-19 pandemic, makes it necessary to improve assistance, services, interventions, and policies so that they can be used without the need for direct contact. A direct example of these activities is enhanced cognitive behavior therapy (CBT-E) via teletherapy [2,50].

The main limitation of our study is that it is a cross-sectional study. Another limitation is that our recruitment method only allowed us to reach volunteers, therefore, we have a limited ability to generalize the results. In addition, we assessed COVID-19-related stress and anxiety with instruments that we used in a Polish sample for the first time. Although efforts were made to assess validity (e.g., factorial validity), more can be done to assess the validity of measures of COVID-19-stress and anxiety in the Polish context. Moreover, the measurement of all variables was based on self-report data (including subjective assessment of body weight), and any changes in lifestyles or eating habits associated with pandemic experience were not collected.

Follow-up surveys would be important to shed more light on COVID-19-related emotions and body weight status in the context of eating disorders and body image and their relations with lifestyle. This may allow us to better explore the mechanisms underlying changes in eating disorders and body image and to identify lifestyle factors that could exacerbate symptoms of eating disorders. Additionally, longitudinal studies would allow for a better understanding of causal relationships, as well as better consideration of some of the limitations of the present cross-sectional design. In particular, such work would benefit from objective measurements of body weight or body composition (e.g., bioelectrical impedance analysis), the use of standardized questionnaires to measure all variables, improved recruitment methods that involve representative Polish samples, and extending the present research to include men. Moreover, the opportunity to offer people suffering from eating disorders evidence-based therapeutic treatment strategies should be a goal for the future. Therefore, it is important to establish whether the currently available interventions would be efficient in reducing COVID-related emotions and eating disorders symptoms and enhancing a positive body image. Moreover, it should also be verified whether, while working on a reduction in emotions related to COVID-19, e-mental-health interventions are as effective as face-to-face interventions. Finally, it should be emphasized that given the rapidly evolving nature of the pandemic, it is essential that we continue research to monitor the negative mental health consequences of the pandemic actively and to provide appropriate support to attenuate increasing eating disorder symptoms.

## Figures and Tables

**Table 1 nutrients-13-01384-t001:** Demographics characteristic of the four clusters.

	Cluster 1 (*N* = 269) Healthy Weight + Low Stress + Low Anxiety	Cluster 2 (*N* = 154) Healthy Weight + High Stress + High Anxiety	Cluster 3 (*N* = 127) Excess Weight + High Stress + High Anxiety	Cluster 4 (*N* = 121) Excess Weight + Low Stress + Low Anxiety	
	*M* (*SD*)				*F*
**Age**	30.30 (10.32) ^a,b^	29.97 (11.06) ^c,d^	36.58 (13.37) ^a,c^	36.33 (9.51) ^b,d^	*F*(3, 667) = 17.05, *p* < 0.001
**BMI**	21.63 (1.72) ^A,B^	21.86 (1.82) ^C,D^	29.96 (4.79) ^A,C^	30.07 (4.40) ^B,D^	*F*(3, 667) = 367.42, *p* < 0.001
	*N* (%)				*χ*^2^ and Cramer’s *V*
**Ethnicity**					𝜒^2^_(6)_ = 9.51, Cramer’s *V* = 0.08 ^NS^
White	267 (99.26)	151 (98.05)	124 (97.64)	121 (100)	
Black	0	0	0	0	
Asian	0	0	0	0	
Mixed race	0	0	2 (1.57)	0	
Other	2 (0.74)	3 (1.95)	1 (0.79)	0	
**Sexual orientation**					𝜒^2^_(15)_ = 11.74, Cramer’s *V* = 0.08 ^NS^
Heterosexual	246 (91.45)	138 (89.61)	115 (90.55)	113 (93.39)	
Lesbian	2 (0.74)	4 (2.60)	5 (3.93)	0	
Bisexual	15 (5.59)	9 (5.84)	4 (3.15)	5 (4.13)	
Pansexual/Queer	2 (0.74)	1 (0.65)	1 (0.79)	1 (0.83)	
Asexual	2 (0.74)	1 (0.65)	1 (0.79)	2 (1.65)	
Other	2 (0.74)	1 (0.65)	1 (0.79)	0	
**Educational qualification**					𝜒^2^_(24)_ = 59.47, Cramer’s *V* = 0.17 ***
Primary school	0	0	0	1 (0.83)	
Intermediate school	0	0	0	0	
Vocational school	0	0	1 (0.79)	0	
Secondary school or technical school	86 (31.97)	49 (31.82)	32 (25.20)	19 (15.70)	
Bachelor or engineer	56 (20.82)	32 (20.78)	19 (14.96)	13 (10.74)	
Master of Arts or Master of Science etc.	76 (28.25)	42 (27.27)	47 (37.00)	55 (45.45)	
Doctor of Philosophy or Doctor of Engineering etc.	2 (0.74)	3 (1.95)	0	1 (0.83)	
Postgraduate studies	29 (10.78)	15 (9.74)	21 (16.54)	29 (23.97)	
Still in full-time education	20 (7.44)	13 (8.44)	6 (4.72)	3 (2.48)	
Other qualification	0	0	1 (0.79)	0	
**Relationship status**					𝜒^2^_(24)_ = 60.39, Cramer’s *V* = 0.17 ***
Single—not currently partnered or married/dating	45 (16.73)	35 (22.73)	23 (18.11)	15 (12.40)	
Partnered—not living with partner	61 (22.68)	41 (26.62)	13 (10.24)	11 (9.09)	
Partnered—living with partner	52 (19.33)	23 (14.94)	19 (14.96)	17 (14.05)	
Married	92 (34.20)	51 (33.11)	62 (48.82)	74 (61.16)	
Divorced	5 (1.86)	2 (1.30)	3 (2.36)	4 (3.30)	
Widowed	2 (0.74)	0	2 (1.58)	0	
In a polyamorous relationship	1 (0.37)	0	0	0	
In an open relationship	3 (1.12)	0	0	0	
Other	8 (2.97)	2 (1.30)	5 (3.93)	0	

^a,b,c,d^ the significant (*p* < 0.001) differences between the groups in terms of age; ^A,B,C,D^ the significant (*p* < 0.001) differences between the groups in terms of BMI; ^NS^ a non-significant; ^***^
*p* < 0.001.

**Table 2 nutrients-13-01384-t002:** Separate univariate ANOVAs on the outcome variables.

	Cluster 1 (*N* = 269) Healthy Weight + Low Stress + Low Anxiety	Cluster 2 (*N* = 154) Healthy Weight + High Stress + High Anxiety	Cluster 3 (*N* = 127) Excess Weight + High Stress + High Anxiety	Cluster 4 (*N* = 121) Excess Weight + Low Stress + Low Anxiety	
	*M* (*SD*)				*Post hoc*
	*F*(3, 667) = 36.84, *p* < 0.001, *ƞ_p_*^2^ = 0.14				
**EDI-BD**	0.57 (0.63)	0.82 (0.73)	1.26 (0.88) ^a^	1.25 (0.83) ^a^	1 vs. 2 ** 1 vs. 3 *** 1 vs. 4 *** 2 vs. 3 *** 2 vs. 4 *** 3 vs. 4
	*F*(3, 667) = 11.37, *p* < 0.001, *ƞ_p_*^2^ = 0.05				
**EDI-DT**	0.52 (0.71) ^a^	0.69 (0.72) ^a,b^	0.97 (0.80) ^c^	0.77 (0.75) ^a,b,c^	1 vs. 2 1 vs. 3 *** 1 vs. 4 * 2 vs. 3 ** 2 vs. 4 3 vs. 4
	*F*(3, 667) = 13.98, *p* < 0.001, *ƞ_p_*^2^ = 0.06				
**EDI-B**	0.31 (0.58) ^a^	0.38 (0.61) ^a,b^	0.73 (0.81) ^c^	0.59 (0.67) ^b,c^	1 vs. 2 1 vs. 3 *** 1 vs. 4 ** 2 vs. 3 *** 2 vs. 4 ^†^ 3 vs. 4
	*F*(3, 667) = 46.59, *p* < 0.001, *ƞ_p_*^2^ = 0.17				
**MBSRQ-AE**	3.58 (0.88) ^a^	3.35 (0.90) ^a^	2.54 (0.94) ^b^	2.81 (0.94) ^b^	1 vs. 2 ^††^ 1 vs. 3 *** 1 vs. 4 *** 2 vs. 3 *** 2 vs. 4 *** 3 vs. 4
	*F*(3, 667) = 16.76, *p* < 0.001, *ƞ_p_*^2^ = 0.07				
**MBSRQ-OWP**	2.22 (0.88)	2.48 (0.93) ^a^	2.78 (0.85) ^b^	2.76 (0.83) ^a^^,b^	1 vs. 2 * 1 vs. 3 *** 1 vs. 4 *** 2 vs. 3 * 2 vs. 4 ^†††^ 3 vs. 4
	*F*(3, 667) = 29.48, *p* < 0.001, *ƞ_p_*^2^ = 0.12				
**MBSRQ-BASS**	3.46 (0.78) ^a^	3.26 (0.79) ^a^	2.75 (0.76) ^b^	2.95 (0.70) ^b^	1 vs. 2 ^††††^ 1 vs. 3 *** 1 vs. 4 *** 2 vs. 3 *** 2 vs. 4 ** 3 vs. 4

EDI—the Eating Disorder Inventory: body dissatisfaction (EDI-BD), drive for thinness (EDI-DT) and bulimia (EDI-B); MBSRQ—the Multidimensional Body–Self Relations Questionnaire: appearance evaluation (MBSRQ-AE), overweight preoccupation (MBSRQ-OWP) and body areas satisfaction (MBSRQ-BASS). ^a,b,c^ If clusters share the same non-capital letter, then the differences between the groups are not statistically significant. * *p* < 0.05, ** *p* < 0.01, *** *p* < 0.001, ^†^
*p* = 0.053, ^††^
*p* = 0.075, ^†††^
*p* = 0.059, ^††††^
*p* = 0.058.

## Data Availability

The data that support the findings of this study are available from the corresponding author upon reasonable request.

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
