# Peer review of "COVID-19-Related Stress and Anxiety, Body Mass Index, Eating Disorder Symptomatology, and Body Image in Women from Poland: A Cluster Analysis Approach"

_nutrients, 2021, doi:10.3390/nu13041384_

Round 1

Reviewer 1 Report

Thank you for gave me the opportunity to read this study.

This work aimed to analyze the presence and severity of some dimensions typically associated to eating disorders in individuals with high stress and anxiety COVID-19 related and excessive body weight. A cluster analysis allowed to classify different condition associated to stress, anxiety and weight status. The analysis reported an association between high levels of anxiety and stress COVID-19 related and high BMI and some variables such as bulimia, overweight preoccupation, body dissatisfaction and drive for thinness.

Although the interesting perspective proposed by this study, some modifications are necessary in order to bring out its application potential and originality.

Abstract: A couple of sentences of rationale are needed to put the work in context. Moreover, the aims should be better expressed.

Introduction:  For an easily access to the related references I suggest including in the list (lines 46-65) references for each point.

The rationale on the association between the variables should be more disclosure. Please, include it before the aims of the study.

Please define after the aims the hypotheses about the expected results according to the literature (a lack of focus emerged in the last part of the introduction).

Discussions: As stated about the introduction, please include studies that have been concerned with investigating the relationship between eating behavior and pandemic, also define how this study fits in the the literature on the topic.

The results should be discussed considering the hypotheses to be included in the introduction.

it is necessary to interpret the results obtained with caution, as they could be determined primarily by body weight rather than pandemic status, since few significant statistics differences emerge between the groups with low and high anxiety and stress covd-19 related.

The limits section should be improved. Some suggestions: any changes in lifestyles or eating habits associated with pandemic experience are collected; use of non-standardized questionnaire to assess covid-19-related anxiety and stress; subjective assessment of body weight.

Please, including a paragraph of conclusion that includes future perspectives and clinical and research implications of this study.

Author Response

Dear Reviewers,

We would like to thank the Reviewers for their careful and thorough reading of this manuscript and for their thoughtful comments and constructive suggestions, which have helped us improve the quality of this manuscript. I've attached our response to review.

Reviewer 2 Report

Section 2.1

Include the ethics approval number here

How was the survey administered and distributed? More information needed.

Section 2.2

How did exactly the translation was carried out? 

Section 2.3

What cluster analysis technique was uused? 

What stats program was used? 

Was the normal 5% sig. level used? If yes, state so

Table 2. Posthoc interaction is hard to read, please assign abc groupings instead beside the mean values in the rows.

Section 3 Results

Only a brief generalised explanation was done here, what is actually interesting here for the results?

Section 4 Discussion 

What is the key results and highlights of the results as mentioned before? Please subsection this part accordingly.

More thorough results discussion from the authors is needed at the current stage the manuscript feels too raw and doesn't highlight the interesting results that the authors have investigated.

What would be most interesting is to create a cause and effect model using say SEM model? Have the authors considered this?

Author Response

(The authors gave the same response as above.)

Round 2

Reviewer 1 Report

I appreciated the work of the authors and I found the work much improved and focused on the most useful aspects of the research compared to the previous version.
I also find the inclusion of the final conclusion paragraph valuable.
I believe that with these improvements the work can be effectively valid for publication, providing interesting insights into the relationship between covid-19 and eating disorders.

Author Response

Dear Reviewer,

Thank you very much.

Reviewer 2 Report

I'd like to thank the authors to take the time for the amendments. 

Table 2. What I'm trying to say is whether the authors can clearly state which post hoc group between Cluster 1-4 where it is significant. 

For example between 1.26 and 0.57 is that significant and grouped them together, refer to the link below

https://stats.stackexchange.com/questions/315687/correct-interpretation-of-anova-post-hoc-results-which-group-is-the-best-one

Author Response

Dear Reviewer,

Thank you very much for your continued efforts to get out manuscript ready for publication. We appreciate the comments concerning Table 2. We hope we were able to improve Table 2 as suggested. I've attached our response.
